# Seed Traits Research Is on the Rise: A Bibliometric Analysis from 1991–2020

**DOI:** 10.3390/plants11152006

**Published:** 2022-07-31

**Authors:** Zhaogang Liu, Ming Zhao, Zhengkuan Lu, Hongxiang Zhang

**Affiliations:** 1Key Laboratory of Ecosystem Network Observation and Modeling, Institute of Geographic Sciences and Natural Resources Research, Chinese Academy of Sciences, Beijing 100101, China; liuzg.20b@igsnrr.ac.cn; 2Northeast Institute of Geography and Agroecology, Chinese Academy of Sciences, Changchun 130102, China; 3College of Resources and Environment, University of Chinese Academy of Sciences, Beijing 100190, China; zhaoming@ibcas.ac.cn (M.Z.); luzhengkuan96@163.com (Z.L.); 4State Key Laboratory of Vegetation & Environmental Change, Institute of Botany, Chinese Academy of Sciences, Beijing 100093, China

**Keywords:** agriculture, bibliometric analysis, ecology, germination, plant traits, seed traits

## Abstract

Seed traits (ST) influence seedling establishment, population dynamics, community composition and ecosystem function and reflect the adaptability of plants and the environmental conditions they experienced. There has been a historical and global accumulation of studies on ST, but with few pertaining to visual and quantitative analyses. To understand the trends in the field of ST research in the past 30 years, we conducted a bibliometric analysis based on the Science Citation Index-Expanded (SCI-E) database. The analysis provided annual publications, time trends for keywords, the most productive journals, authors, institutions and countries, and a comprehensive overview of the ST field. Our results showed that in the past 30 years, the number of publications in ST research has increased at an average annual growth rate of 9.1%, while the average number of citations per paper per year showed a rapid increase–slow increase–decrease trend. Keyword analysis showed that “germination” was the most popular research section. *Crop Science* ranked first among the top journals and *Theoretical and Applied Genetics* had greater influence in this area and more citations than other journals. The 10 most productive institutions were mostly located in the United States, China and Australia. Furthermore, the three countries also had the largest number of publications and citations. Our analysis showed that the research interests in ST have evolved from genetics and agricultural science to ecological research over the last thirty years; as more fields embrace ST research, there are opportunities for international and interdisciplinary collaborations, cooperative institutions and new advances in the field.

## 1. Introduction

Plant traits, also known as plant functional traits, are usually measurable traits that are closely related to plant production optimization and environmental adaptation through long-term adaptation and co-evolution [1]. For example, specific leaf area (SLA) refers to the leaf area per unit dry weight, which is one of the most widely studied plant functional traits in ecology. It has been demonstrated that SLA has a great impact on plant adaptability, plant growth and photosynthetic rate [2,3]. The method based on plant traits has further improved our understanding and cognition of plant evolution, community composition and ecosystem functions at different levels [4,5,6,7]. Despite the immense popularity of trait-based frameworks to explain plant functioning and performance, seed traits (ST) are a relatively recent focus of the field.

Seed traits (ST), such as seed size, shape, surface and its appendages, color, seed coat thickness, storage reserves and dormancy type, not only express a certain amount of relatively stable genetic information, but are also closely related to seed production, dispersal, predation, persistence in the soils, germination and other processes [8,9,10,11,12]. Seeds store energy and nutrients to support the initial growth and development for plants, meanwhile, it increases offspring fitness [13], enabling the population to survive during adverse periods and colonize after disturbance [14]. A series of morphological and physiological characteristics enables seeds to coordinate germination time under appropriate conditions to establish seedlings. The characteristics of seeds and fruits enable them to be transmitted through animals, humans, wind and water. Seeds may spread to more places and be exposed to a wider range of environmental conditions for a longer period of time, thus increasing the success rate of plant reproduction and the possibility of subsequent diversification and improvement of local adaptability. The ability to disperse in space and time has a profound impact on the genetic diversity of plants, so it also affects the adaptive dynamics [7]. The characteristics of seeds are one of the most important issues to understand the evolutionary ecology of plants, and how global climate change will ultimately affect plants and ecosystems [7].

ST have received less attention in plant science than plant traits, such as leaf traits and root traits [3,15], except for seed size and mass [16,17]. More and more studies have shown that the inclusion of ST in the study of community ecology is helpful to better understand community assembly [18,19,20,21,22] and to clarify the colonization patterns and species persistence in the community over time [23]. In light of the importance of seed traits at natural population, community and ecosystem scales and in agriculture, a growing number of related studies has been conducted in recent decades [24]. For example, Saatkamp et al. (2019) have proposed a seed-trait functional network, which provides novel insights and can be incorporated into evolutionary ecology, community ecology and applied ecology.

In the field of seed research, Shi, Zhang and Wei (2020) analyzed the progress in soil seed banks’ research by using CiteSpace, based on the publications from 1900 to 2019 from the Web of Science Core Collection database [25]. Morales, et al. [26] conducted a bibliometric analysis of the scientific production and citations of the International Maize and Wheat Improvement Center from the Web of Science citation indices, where the emphasis was placed on the development of plant genetics to improve the yield and quality of new seed varieties. However, to the best of our knowledge, no specific bibliometric analysis has been conducted on the whole of ST studies. Thus, this study aims to use bibliometrics to analyze the trends and key areas of ST research from 1991 to 2020. The results will help researchers to evaluate trends, gaps and research directions in seed-related fields. Specifically, we aim to: (1) provide an accurate overview of the scientific publications over time; (2) understand the publishing patterns of studies globally in the ST field; and (3) develop future strategies for research in this area.

## 2. Data Analysis and Methods

### 2.1. Data Collection and Preparation

Bibliometric analysis is a statistical method that uses modern technologies, such as computer engineering, database management and statistics, to quantitatively evaluate articles, books and any other publications. It has been applied in a lot of fields of study [27,28,29]. Although bibliometrics are quantitative in nature, these methods are also used to illustrate qualitative features. Therefore, the bibliometric analysis of the progress in specific scientific research fields is called scientific science. Although the response of this analysis method to the latest research is weak, it helps new researchers find research trends and hot spots and obtain major classic publications [29].

Our research mainly focused on ST. In this part, data collection and preparation were mainly divided into two stages. The first stage was the data retrieval. From 1991 to 2020, all of the publications were searched with “seed traits” as the subject. We choose 1991–2020 as the research period, because there were few pieces of research before 1991 in the database. We started our bibliometric collection work on 8 February 2021 accessing the Science Citation Index Expanded (SCI-E) database in the “Web of Science Core Collection” (Web of Science at http://www.webofknowledge.com). The SCI-E database comprehensively covers the most influential research in the world and contains clear reference information, enabling us to track the trends in ST research [29]. We preliminarily screened the publications, mainly including 11 aspects (such as publication type, keywords, correspondence, title, abstract, number of citations, year, month/day, volume, issue and digital object identifier). For example, several pre-processing methods were applied to detect the repeated and misspelled elements. Although most of the bibliometric data are reliable, the references cited can contain multiple versions of the same publication and different spellings of the author’s name. In addition, authors usually use their last names and initials, so common names can be problematic. The cited periodicals can also appear in slightly different forms. In our analysis, there were 24,090 documents, mainly including papers, articles, reviews and letters (see Table 1 for more details). Downloading and converting the data were the second stage of our process. For the following bibliometric analysis, we converted the data into BibTex format (Bibliometrix package in R software).

### 2.2. Statistical Analysis

This section provides a general statistical analysis, such as the number of publications and the number cited by each country, journal, author and institution.

Research trends and frontiers can be well reflected by frequently used keywords [27]. Wordcloud can quickly present the most important words and locate different words in alphabetical order to determine their relative importance in related fields. We used wordcloud to identify the 50 most frequently used keywords in ST studies over the past 30 years. Each tag represents a single phrase, and different font sizes and colors represent different frequencies [30].

We produced an author collaboration and a country collaboration network. The size of the circle represents the production of the author/country. The larger the circle is, the higher production rate the author/country has. The lines between two circles indicate the relationships between two authors/countries. The thicker the line is, the closer the relationship the two authors/countries have. Different colors represent different clusters, indicating that these authors/countries are more likely to appear in the same publication [30].

The Bibliometrix R-package allows multiple correspondence analysis (MCA) to be performed, using the conceptual structure function to plot the conceptual structure of the fields, and K-means clustering is used to identify the document clusters that express common concepts. The results are interpreted based on the relative position of the points and their distribution along the dimensions. The more closely related the keywords are, the more they will be in a cluster or grouping.

In order to better understand the time evolution of the studied topic, we built a time-trend evolution analysis of the keywords and divided publications in the ST fields for 30 years into six periods (1991–1995, 1996–2000, 2001–2005, 2006–2010, 2011–2015 and 2016–2020). By dividing the 30-year time span into different time periods, alluvial maps can be used to represent the time evolution of topics in a specific research field [30]. All of the keywords form a cluster and each cluster contains several keywords. These keywords are actually several main keywords in the keyword cluster and all of the clusters appear in the timeline according to the sequence number. This timeline chart just spreads out the keywords containing in the cluster according to the year of their appearance. Thus, the timeline map is generated

In this paper, all of the bibliometric statistics and text analysis (e.g., number of annual publications and citations, most productive journals, most productive authors, most productive institutions, most productive countries and most used keywords), data analysis and mapping were carried out using R 4.0.3 software (Bibliometrix package in R software) and SigmaPlot 12.5 software.

## 3. Results and Discussion

### 3.1. Temporal Trends of Publications and Citations

The annual circulation of publications can reflect the research progress and its importance of ST to some extent. The overall trend of the literature showed that the number of publications increased during 1991–2020 (Figure 1a). We also found that the total number of research papers on ST was very large, which indicates that ST research expanded during the past thirty years. The number of studies increased nearly 12 times, from 178 in 1991 to 2201 in 2020, with an average annual growth rate of 9.1%. After 2001, there was a significant increase, and nearly 90% of the publications were published between 2001 to 2020. The increase trend of ST-related publications was consistent with that of the global scientific publications [29]. On the other hand, ST have received less attention historically and are now growing in attention, and there are still many unsolved scientific problems.

Although the number of publications has been increasing in the past 30 years, the average number of citations per paper per year showed a rapid increase–slow increase–decrease trend (Figure 1b). During 1991–2000, it was in the stage of rapid increase, during 2000–2015, it was in the stage of steady growth, and during 2015–2020, it was in the stage of decline. One possible reason is that new publications will take a longer time to reach their peak, and old publications will gradually lose their status because they will be replaced by new publications. Another reason why the average number of citations per paper per year is unstable may be related to the number of publications published in high impact journals each year [31]. Researchers always prefer to cite papers from high impact journals. For example, the number of publications in 1996 was the same as that in 1994, but the citations were higher than that in 1994, owing to the higher number of publications in high impact factor journals (Figure 1b). Therefore, ST-related researchers should try to publish their work in high impact factor journals in the future to improve the citations and influence of their work.

### 3.2. Related Journals

At present, the ST studies have appeared in 1598 journals. This reveals that ST-related publications were highly dispersed in various journals. As shown in Table 2, *Crop Science*, *Euphytica*, *Theoretical and Applied Genetics*, *PLoS ONE* and *Frontiers in Plant Science* were the five most productive journals for ST studies. These journals are professional journals or comprehensive journals that are closely related to plants (crops).

Citation statistics are often used to assess the relative impact of academic journals. *Theoretical and Applied Genetics*, *Crop Science*, *Ecology*, *Plant Physiology*, and *PNAS* were the five most cited journals for ST studies (Table 2). These analyses showed that the journals with more ST research papers published tended to have higher citations of ST studies, e.g., *Crop Science*. In addition, the journals with high impact factors tended to have higher citations of ST studies, although they had few publications in the ST research field, e.g., *Ecology*.

In our study, we found that the top five productive journals in ST-related studies showed different time trends in the 30-year development process (Appendix A). In recent years, the open access journals, *PLoS One* and *Frontiers in Plant Science* maintained a dramatic growth trend in the publication of ST studies, while publications of ST studies in the traditional journals *Euphytica*, *Theoretical and Applied Genetics* and *Crop Science* slowly increased.

### 3.3. Productive Authors

Through the analysis of the authors, we identified researchers who engaged in ST-related research and made important contributions. Ten high-yielding authors published 1612 papers, accounting for 6.7% of the total publications. Appendix A shows the top 10 most productive authors and the top 10 most cited authors. It should be noted that, when calculating the publications and citations, we do not distinguish the order in the author list. In other words, as long as the name was listed in the author list, we will record it.

Figure 2 shows the collaboration network of the productive authors. The size of the circle indicates the number of publications published by the author, and the connection between authors indicates the intensity of cooperation. By analyzing the cooperation network of the high-yield authors, we can quickly grasp the main research teams in the field. It is not difficult to find that Chinese authors rarely cooperate with national and international scientists, as they are divided into several teams. This may be one of the reasons why none of these authors were the most cited authors. Wide collaboration is an effective way to advance the development of science.

### 3.4. Productive Institutions and Countries

The marked institution can reflect the academic attention in ST research and help identify the activity and the influential institutions. Through this research, 11,701 institutions around the world involved in ST research, and the 10 most marked institutions contributed 5106 papers, accounting for more than 20% of the total number of searched publications. In the past 30 years, the University of Western Australia ranked first with the largest number of papers (Table 3). Huazhong Agricultural University and Nanjing Agricultural University ranked second and third, respectively. It should be noted that most of the top ten productive institutions were located in the United States, China and Australia, which may be because they are three typical arid and semi-arid regions in the world [32]. They need to increase seed yields to meet the increasing population and minimize the adverse effects of land decline.

The productive countries may reflect the importance that the country gives to ST research. To some extent, the number of papers published represents the development level of the country in this study area. We created a world heat map showing publications across all of the countries (Figure 3). This could help visualize the larger regions that have the most productive or cited institutions and where research hot spots occur. According to our study, 123 countries in the world have engaged in ST-related studies (Figure 3). The 10 most relevant countries by corresponding author contributed 15,583 papers (64.7%) and the 10 most total cited countries contributed 399,999 total citations (72.9%) (Appendix A). During the study period, USA, China, India, Brazil, Canada, Germany, Spain, Australia, Japan and France were the 10 most productive countries by corresponding authored publications. However, USA, China, Australia, Germany, UK, Canada, Spain, France, India and Japan were the 10 most cited countries (Appendix A). The cooperation between the USA, China and Australia was relatively close. However, these three countries were less cooperative with the European countries, which needs to be strengthened in future research (Figure 4). Although China ranked second in the number of papers and citations, there was still a big gap between China and the USA. This analysis enables us to identify the research differences in the number of ST papers and citations in many countries and emphasize the need to develop ST research and promote cooperation among researchers around the world.

### 3.5. Temporal Evolution of Popular Keywords

#### 3.5.1. Most Popular Keywords

Keywords show information about research trends and frontiers, as well as the topics of most interest to researchers in the field [29,33,34]. From a total of 40,057 keywords, wordcloud showed the 50 most used keywords in the past 30 years since 1991 (Figure 5). The most frequently used keywords were “germination” or “seed germination” (total frequency of 804), except for our search terms “seed”. Seed germination is the most important life-cycle stage affecting population establishment and regeneration. It is not only affected by temperature, light, soil moisture and other environmental factors, but is also closely related to the characteristics of the parents and ST [33,34,35,36,37]. Many researchers in the fields of seed science, agriculture, botany and ecology have expressed concerns about seed germination. Therefore, the research related to germination has always been an important field of seed science studies.

Other keywords that play an important role in ST research include “yield”, accompanied by some crops, and “heritability” (Figure 5). The genotype differences in seed yield and its components among different families are very large, indicating that there is great potential to improve these traits by directional selection in breeding programs [38]. Many agronomic traits (such as seed yield) used for crop improvement are highly complex quantitative traits controlled by many genetic loci, which poses a challenge to comprehensively capture the relevant markers/genes [39]. For this purpose, scientists are making a lot of effort in researching the relationship between yield and heritability in the ST research fields.

We found that seed size, seed dispersal, seed mass and seed dormancy were the most studied ST, besides seed germination, which are the core of the seeds’ functional traits (Figure 6). Seed size/mass is one of the key traits that affect the ability of seed dispersal and animal-mediated seed transmission [40]. Seed mass and shape are the traits most closely related to seed persistence and seedling vigor in the soil seed bank [8,9,10]. Other ST, such as germination timing, can show how species share resources and spread risks in the early stages of their development [11]. Seed dormancy prevents germination when conditions are not favorable for subsequent seedling survival and establishment [41]. In total, we found that, at present, there were mainly two aspects in this field, as shown in Figure 6, mainly focusing on the research of seed traits in natural ecosystems and genetics in agriculture.

Seed ecology can be defined as the science of studying the life cycle, character adaptation and evolution of seeds under the influence of the biotic and abiotic environments. The possible processes and temporal and spatial changes of seeds from “birth” to “death” are the main research contents of ST-related studies, including seed development, seed dispersal, seed predation, soil seed bank, seed dormancy and germination, seed character adaptation and evolution. Each process is faced with the risk of death (Figure 7). In order to guide the data collection and empirical work of incorporating ST into functional ecology, we suggest that key traits should be related to four key seed functions: dispersal; persistence; germination timing; and establishment (Figure 7). The four functions capture the important role of seeds in population, community and ecosystems, determining the direction of the relevant characteristics and their trade-offs in the regeneration process. A key set of ST that are closely related to these specific functions was also proposed (Figure 7). Only a few ST have been successfully incorporated into the mechanism framework, and most ST still need to empirically test their hypothetical functions. 

ST should also be analyzed together with other plant functional traits to fully understand their interactions, such as the relationship between plant height and dispersal, or the relationship between life cycle strategies and seedling establishment. Our understanding of plant plasticity, adaptability, evolution, distribution and dynamics may need to deepen our cognition of ST and function, so as to provide some theoretical basis for predicting how human activities and climate change influence plants and natural ecosystems. As long as we identified ST and the functions related to seed dispersal, persistence, germination time and establishment, ecologists can integrate these traits into our cognition of the ecological and evolutionary processes at the organizational and spatiotemporal dimensions. In order to improve the long-term protection and recovery of cross-generational success, we advocate strengthening the research on the four seed functions in application. These studies will facilitate the transition from restoration strategies for degraded or specific ecosystems to a broad understanding of the characteristics that guide recovery strategies and improve their success rates.

**Figure 7 plants-11-02006-f007:**
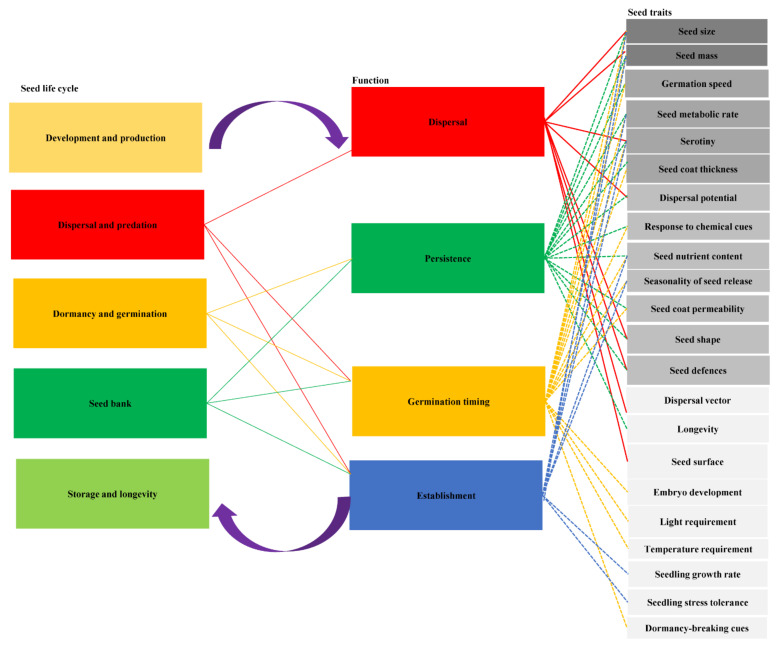
Conceptual graph of network between seed life cycle, seed functions and seed traits. Revised from Saatkamp et al. 2019 [7].

#### 3.5.2. Temporal Evolution of Keyword Frequencies

The time span is divided into different time segments, and the evolution trend of topics in a specific study area can be represented by the alluvial graph [30]. The time evolution of all of the keywords in ST research showed that the most used keywords have changed significantly in the research process. “Germination” was a research hotspot in the 1990s, while “yield” was a research hotspot in the 2010s. The popular keyword “heritability” flourished between 1991–2010 and then evolved in three main directions: gene flow; yield; and genetic diversity.

From 1991 to 1995, “germination”, “seed”, “plant”, “soybean” and “seed dispersal” were the main research keywords in classical agriculture, and “heritability”, “heterosis” and “selection” became the hot topics in genetics (Figure 8). With the development of basic agriculture and genetics’ studies, practical research increased between 1995 and 2000, especially in “rice”, “pollination”, and “phaseolus vulgaris” (Figure 8). During 2001–2010, the research subjects were similar to the previous topics. After 2011, “functional traits”, “genetic diversity”, “gene flow” and “yield” became new popular fields of research (Figure 8).

The transformations of the most-used keywords showed the changes in the core research fields of ST in past 30 years. In the 1990s and the 2000s, ST researchers mainly studied classical genetics and agriculture studies, and in the 2010s, they mainly studied ecology-based ST. This analysis is useful as it provides valuable information and potential new research trends for ST researchers and help them to identify proper research topics. For example, in the 1990s, with the policies of “moving towards science” and “spring of science”, most of the Chinese scientists devoted themselves to basic studies [42]. At the beginning of the new millennium, scientists started to concern themselves with land degradation, poverty and biodiversity protection [43]. In the context of current global climate change, there is an urgent need to better understand the impact of climate change on ST, which is also an important and difficult topic.

## 4. Conclusions and Limitations

We presented a comprehensive overview of published articles between 1991 and 2020 in the seed traits research field, based on the bibliometric analysis. The research on seed traits has increased dramatically, but the average number of citations per paper per year showed a rapid increase–slow increase–decrease trend. The most frequently used keywords were ‘‘germination’’, “yield” and “heritability” in seed traits research.

During the recent three decades, *Crop Science*, *Euphytica*, *Theoretical and Applied Genetics*, *PLoS ONE* and *Frontiers in Plant Science* were the five most productive journals. However, *Theoretical and Applied Genetics*, *Crop Science*, *Ecology*, *Plant Physiology*, and *PNAS* were the five most cited journals. The 10 most productive institutions were mostly located in the United States, China and Australia and they also had the greatest research publications and most citations. These results showed that the quality of the published papers was far more important than the total number of the published papers. We built the studies trend of seed traits and the time evolution of popular keywords, and revealed that classical genetics and agricultural science were the main topics in the 1990s and 2000s, and there was a trend towards ecological research in the 2010s.

Our analysis of this paper was carried out by the SCI-E database. The articles published in non-SCI journals and other languages were not listed in the database, which may lead to insufficient coverage of the study. It should be supplemented by searching other databases in the future. Although the scope of search results was limited, our studies provide a good starting point for elucidating the time evolution of seed traits. Our findings indicate that the research on seed traits are limited in space, quantity and quality. More large scale studies and influential studies on seed traits are needed, e.g., the responses of seed traits to climate change. Therefore, it is necessary to further develop studies and promote cooperation, which requires funding and international coordination among scientists from different perspectives and disciplines.

## Figures and Tables

**Figure 1 plants-11-02006-f001:**
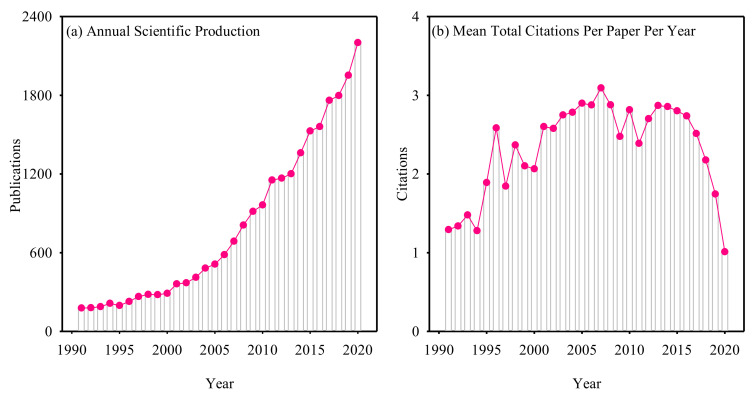
Annual scientific production of publications on seed traits study (**a**) and mean number of citations per paper per year (**b**) from 1991 to 2020.

**Figure 2 plants-11-02006-f002:**
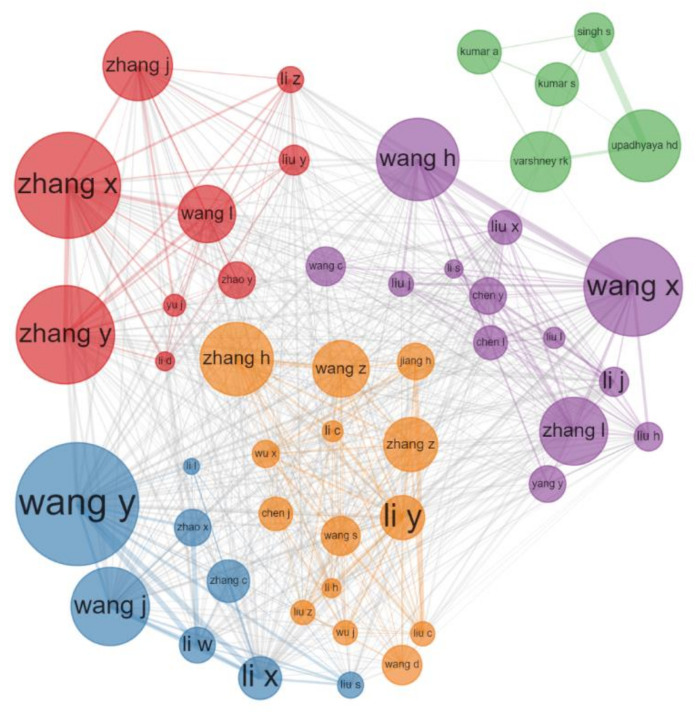
Author collaboration network.

**Figure 3 plants-11-02006-f003:**
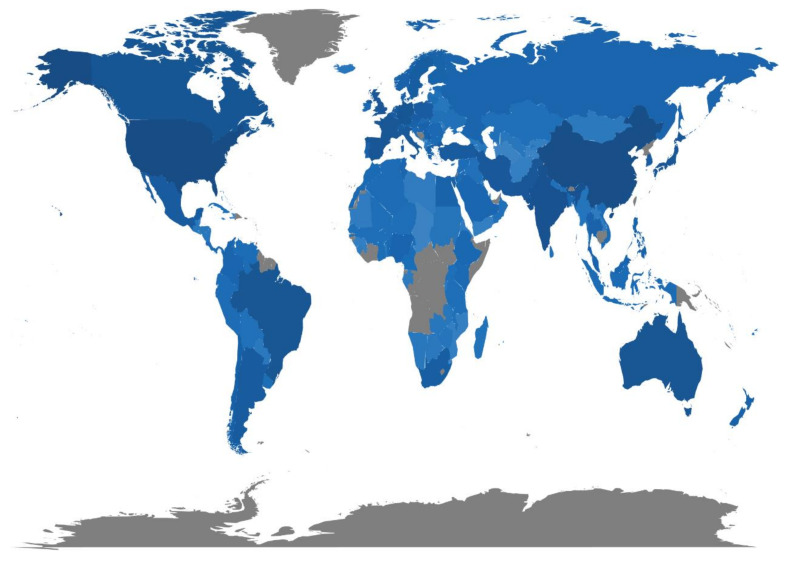
Country scientific production. The color intensity is proportional to the number of publications in ST studies.

**Figure 4 plants-11-02006-f004:**
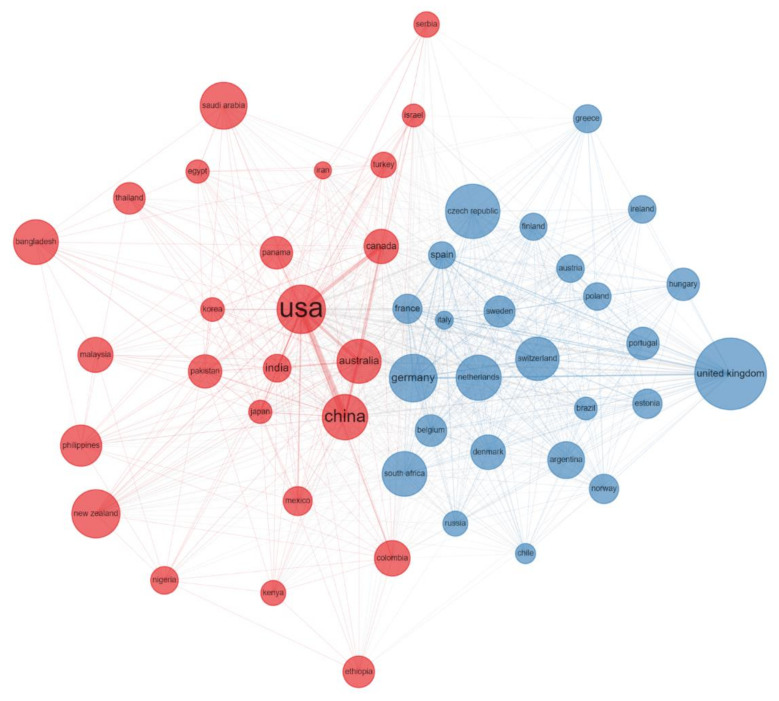
Country collaboration network.

**Figure 5 plants-11-02006-f005:**
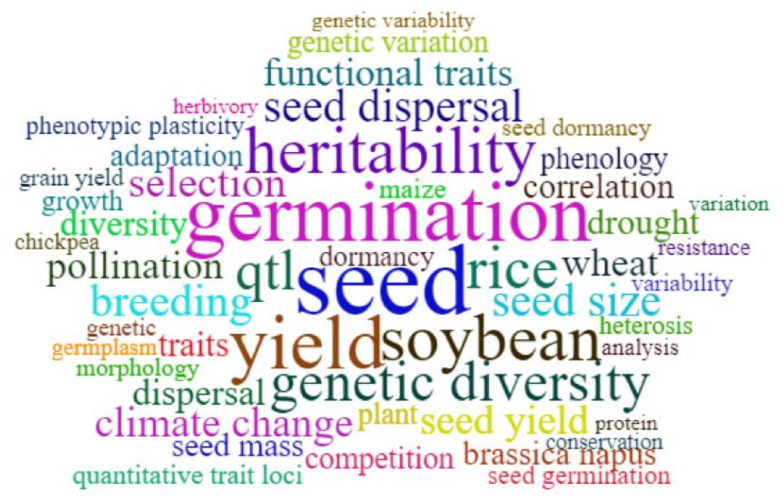
Top 50 frequently used keywords represented by the wordcloud. Labels are usually single words, and the size and color of labels represent different frequencies.

**Figure 6 plants-11-02006-f006:**
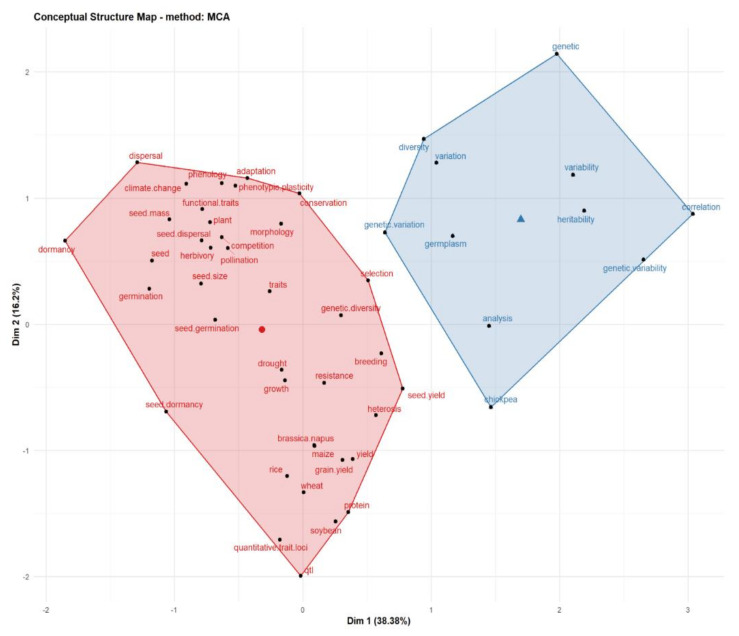
Conceptual map and keywords clusters.

**Figure 8 plants-11-02006-f008:**
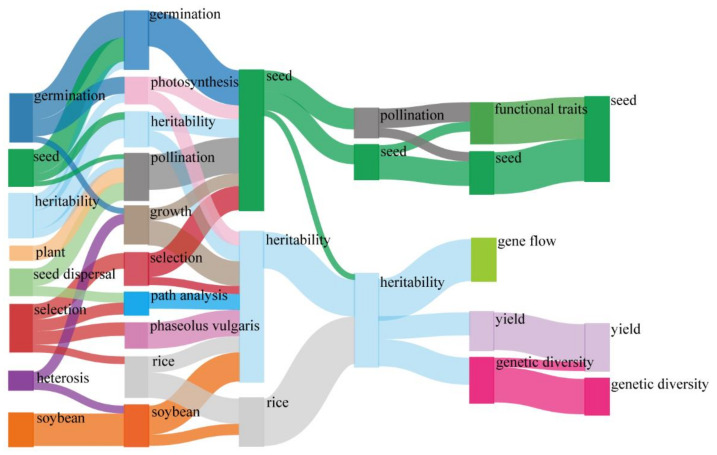
The temporal evolution of popular keywords on seed traits (ST) study. The horizontal axis represents the year. Each node represents a popular keyword, and the size of each node is proportional to its reference frequency. The lines between each node represent the time evolution of keywords. These lines reflect the transfer between keywords and the relationship between heritability. Different colors represent different keywords.

**Table 1 plants-11-02006-t001:** Main information regarding the data collection of seed traits (ST).

Descriptions	Results
Time span	1991–2020
Sources (journals, books, etc.)	1598
Publications	24,090
Keywords	40,057
Authors	57,845
Average publications per year	9.25
Average citations per publication	22.78
Author appearances	111,468
Number of single author publications	1046
Number of multi-author publications	23,044
Publications per Author	0.416
Authors per publication	2.4
Co-authors per publication	4.63
Collaboration Index	2.5

Note: Collaboration index = Authors of multi-authored publications/Multi-authored publications.

**Table 2 plants-11-02006-t002:** Top 20 most marked and cited journals with the publications of seed traits (ST) during the period of 1991–2020.

Journal	Publications	Journal	Total Citations
*Crop Science*	805	*Theoretical and Applied Genetics*	31,680
*Euphytica*	735	*Crop Science*	30,163
*Theoretical and Applied Genetics*	622	*Ecology*	22,440
*PLoS One*	499	*Plant Physiology*	17,843
*Frontiers in Plant Science*	490	*Proceedings of the National Academy of the* *Sciences of the United States of America*	17,523
*Plant Breeding*	330	*Journal of Ecology*	17,279
*American Journal of Botany*	286	*Evolution*	16,661
*Pakistan Journal of Botany*	279	*Euphytica*	14,391
*Journal of Ecology*	278	*American Journal of Botany*	14,353
*Genetic Resources and Crop Evolution*	270	*Oecologia*	13,747
*Annals of Botany*	259	*New Phytologist*	13,563
*Field Crops Research*	254	*American Naturalist*	13,556
*Molecular Breeding*	245	*Genetics*	13,001
*Oecologia*	239	*Science*	12,807
*Legume Research*	231	*Nature*	12,128
*Ecology*	217	*Plant Cell*	11,170
*Scientific Reports*	214	*Oikos*	11,051
*New Phytologist*	210	*PLoS One*	10,621
*Indian Journal of Agricultural Sciences*	199	*Annals of Botany*	10,218
*Industrial Crops and Products*	198	*Journal of Experimental Botany*	10,018

**Table 3 plants-11-02006-t003:** Top ten most marked institutions.

Institution	Publications
University of Western Australia	637
Huazhong Agricultural University	579
Nanjing Agricultural University	553
University of California Davis	539
International Crops Research Institute for the Semi-Arid Tropics	520
The University of Georgia	473
Institute of Botany, the Chinese Academy of Sciences	468
University of Agriculture, Faisalabad	463
University of Minnesota	453
Islamic Azad University	421

## Data Availability

Upon a reasonable request, the authors provided data that supported the findings of this study. This article, along with any supplementary files, includes the entire dataset supporting the findings of this study.

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
