# Peer review of "Seed Traits Research Is on the Rise: A Bibliometric Analysis from 1991–2020"

_plants, 2022, doi:10.3390/plants11152006_

Round 1

Reviewer 1 Report

Comments attached.

Reviewer 2 Report

In this manuscript, the authors described a research trend in the seed traits area using the bibliometric analysis during 1991–2020. This approach is interesting and has a particular contribution to the field. The manuscript is well discussed, and I recommend accepting it after minor revision. 

Minor points:

 1) This work is about seed trait. Please give a short definition of this term in the Introduction. 

2) Please provide a full list of keywords used for analyses. It can be done as a supplement.

3) I was surprised that the top 20 most cited journals (table 2) do not include specialized seed journals such as Seed Science Research, Seed Science and Technology. Please provide a full list of cited journals with the publications of seed traits. It can be done as a supplement.

4) In Figure 6, the quality needs to be improved. Please increase the fonts.
